

# Natural language processing for analyzing online customer reviews: a survey, taxonomy, and open research challenges

Nadia Malik[1],* and Muhammad Bilal[2,3,*]

[1] Department of Management Sciences, COMSATS University Islamabad, Islamabad, Pakistan
[2] Department of Computing and Information Systems, School of Engineering and Technology, Sunway University, Petaling Jaya, Selangor, Malaysia
[3] Department of Pharmaceutical Outcomes and Policy, Malachowsky Hall for Data Science and Information Technology, University of Florida, Gainesville, Florida, United States
* These authors contributed equally to this work.

Corresponding author
Muhammad Bilal,
mbilal.csit@gmail.com

## ABSTRACT

In recent years, e-commerce platforms have become popular and transformed the way people buy and sell goods. People are rapidly adopting Internet shopping due to the convenience of purchasing from the comfort of their homes. Online review sites allow customers to share their thoughts on products and services. Customers and businesses increasingly rely on online reviews to assess and improve the quality of products. Existing literature uses natural language processing (NLP) to analyze customer reviews for different applications. Due to the growing importance of NLP for online customer reviews, this study attempts to provide a taxonomy of NLP applications based on existing literature. This study also examined emerging methods, data sources, and research challenges by reviewing 154 publications from 2013 to 2023 that explore state-of-the-art approaches for diverse applications. Based on existing research, the taxonomy of applications divides literature into five categories: sentiment analysis and opinion mining, review analysis and management, customer experience and satisfaction, user profiling, and marketing and reputation management. It is interesting to note that the majority of existing research relies on Amazon user reviews. Additionally, recent research has encouraged the use of advanced techniques like bidirectional encoder representations from transformers (BERT), long short-term memory (LSTM), and ensemble classifiers. The rising number of articles published each year indicates increasing interest of researchers and continued growth. This survey also addresses open issues, providing future directions in analyzing online customer reviews.

## INTRODUCTION

In the digital era, online customer reviews influence consumer choices. Customers seek peer reviews for assurance and guidance in a global market with alternatives. Reviews offer real-world insights into product and service pros and cons, going beyond marketing narratives (*Dwivedi et al., 2021*). Online reviews affect e-commerce sites like Amazon and

Flipkart and hospitality platforms like TripAdvisor and Yelp. The influence of online customer reviews goes beyond consumers. Businesses that actively engage with internet reviews can learn about the strengths and weaknesses of products. Customer review openness and genuineness generate a sense of community among customers. Online customer review analysis is essential for businesses that want to be competitive, responsive, and customer-centric in the digital marketplace (*Thakur, 2018*).

Businesses looking to improve their goods, services, and customer satisfaction must understand the sentiments and opinions of these reviews. Obtaining valuable insights from online reviews is difficult due to the large number of daily reviews across platforms (*Sezgen, Mason & Mayer, 2019*). Natural language processing (NLP) uses methods and techniques to process human-readable text. In online customer review analysis, NLP extracts attitudes, views, and topics from textual data. Sentiment classification groups review as positive, negative, or neutral, giving a broad snapshot of customer opinion. Opinion mining goes further by finding and categorizing particular thoughts in reviews, providing a more detailed insight into customer feedback (*Sun, Luo & Chen, 2017*).

NLP's capabilities go beyond sentiment analysis and opinion mining. Aspect-based sentiment analysis extracts sentiments about specific product or service attributes, giving businesses actionable recommendations for improvement (*Chauhan et al., 2023*). Emotion analysis adds to customer sentiment by capturing review emotions. As NLP advances, academics and companies may utilize topic modeling, summarization, and deep learning to gain insights from customer reviews' diverse and complicated language. Using NLP to analyze online customer reviews is a technological innovation and a strategic need for organizations seeking digital competitiveness (*Gupta & Patel, 2021*; *Mehra, 2023*).

The textual nature and huge volume of online customer reviews make handling, understanding, and analyzing customers' opinions and feedback challenging. This survey aims to provide a holistic view of the existing state-of-the-art literature on analyzing online customer reviews using various NLP techniques. Several surveys and reviews have been published in recent years that examine the application of NLP to analyze online customer reviews. However, the scope was limited to a specific application area, such as sentiment analysis (*Jain, Pamula, Srivastava 2021*), opinion mining (*Subhashini et al., 2021*), text summarization (*Boorugu & Ramesh, 2020*), automated responses (*Olujimi & Ade-Ibijola, 2023*), and review helpfulness (*Bilal et al., 2019*; *Saumya, Roy & Singh, 2023*). This survey attempts to give a broader overview of the domain and categorize the existing literature based on application area to propose a taxonomy of applications of NLP-based analysis of online customer reviews. It also gives readers an in-depth understanding of the methodologies, tools, and datasets utilized and the open research challenges in this growing field to enable future research and developments.

The potential audience of this survey includes researchers, businesses, and technologists. This survey provides an overview of existing state-of-the-art techniques and open research challenges from the perspective of researchers interested in this domain. From the perspective of businesses, this survey helps identify potential ways in which online customer reviews can be used to boost their business performance. This survey also influences how businesses understand, respond to, and utilize online customer reviews. It

serves as a guide for the technologist working with the online platforms in applying innovative solutions to improve these platforms and overcome information overload. Overall, this survey helps academics, practitioners, and policymakers navigate the complex domain of online customer review analysis.

The remainder of the article is divided into sections. 'Survey Methodology' discusses the method of selecting the articles and research questions included in this survey. 'Taxonomy of NLP Applications in Online Customer Reviews' defines the taxonomy and categorizes the literature by application area. 'Discussion' provides an overview of application areas, data sources, techniques, and open challenges and future directions. Finally, the findings of the study are summarized in 'Conclusion.'

## SURVEY METHODOLOGY

A carefully formulated search query was used to include research on online reviews, customer feedback, and NLP applications in electronic commerce. The search query included several relevant keywords and phrases to cover the topic thoroughly. The search query used was arranged as (TITLE-ABS-KEY(("Online Reviews" OR "Customer Reviews" OR "App Reviews" OR "Google Apps Reviews" OR "NLP for Online Reviews" OR "Sentiment Analysis" OR "Review Summarization" OR "Opinion Mining" OR "Aspect-Based Sentiment Analysis" OR "Named Entity Recognition" OR "Emotion Analysis" OR "User Profiling" OR "Anomaly Detection" OR "Design" OR "Defects" OR "Quality" OR "Comparison" OR "Sale" OR "Reputation" OR "Helpfulness Prediction" OR "Ranking" OR "Rating Prediction" OR "Fake Reviews") AND ("Yelp" OR "Amazon" OR "Google Reviews" OR "TripAdvisor" OR "Trustpilot" OR "App Store Reviews" OR "Play Store Reviews") AND ("Natural Language Processing" OR "NLP") AND ("Electronic Commerce" OR "E-Commerce" OR "M-Commerce" OR "E-Markets" OR "Electronic Business" OR "E-Business") AND PUBYEAR > 2012 AND PUBYEAR < 2024 AND (LIMIT-TO (DOCTYPE, "ch") OR LIMIT-TO (DOCTYPE, "ar") OR LIMIT-TO (DOCTYPE, "cp")) AND (LIMIT-TO (LANGUAGE, "English"))). The names of popular online review platforms were also included in the search query, which aims to ensure the retrieval of articles directly related to the use of data from popular online review platforms. The inclusion and exclusion criteria are provided in Table S1. The literature was filtered for relevance and quality using inclusion and exclusion criteria. First, studies are required to deal with NLP and online reviews in e-commerce. Conference papers, articles, and chapters published from 2013 and 2023 in English were included. Studies that did not meet the publication type, language criteria and the study area were excluded. Portions of this text were previously published as part of a preprint (https://www.preprints.org/manuscript/202312.2210/v2) (*Malik & Bilal, 2023*).

Figure S1 depicts a thorough screening and relevance assessment of each retrieved article. A comprehensive title and abstract screening was performed on 1,256 identified articles from scientific databases, including Scopus, ACM Digital Library, and IEEE Xplore. In this phase, articles that did not meet research objectives were removed. After thorough screening, 473 records were retained for further analysis. The articles that were kept went through a more thorough full-text examination afterward. This crucial step

included a detailed analysis to understand each article's contribution. This comprehensive screening included articles contributing substantially to NLP and online customer reviews. Following the full-text examination, 154 articles were chosen that satisfied the inclusion criteria. Relevant data points from chosen publications were methodically retrieved and categorized. Key results, methodology, and overall themes from each study were identified. Extracted data served as the foundation for later analysis and synthesis of the literature, contributing to the overall goals of this survey.

The following research questions are used in this survey for data extraction and analysis of existing studies:

RQ1: What are the application areas for analyzing online customer reviews?

RQ2: Which data sources of online customer reviews were utilized?

RQ3: What techniques and methods are used to analyze online customer reviews?

RQ4: What are open research challenges and future directions in analyzing online customer reviews?

## Taxonomy of NLP applications in online customer reviews

The taxonomy presented in this section is derived from an extensive literature review. This survey systematically tabulated all selected articles and categorized them based on their content to identify different application areas. The aim was to construct a comprehensive taxonomy that accurately reflects the landscape of NLP applications in the context of online customer reviews. This section covers the taxonomy of NLP applications, including sentiment analysis and opinion mining, review analysis and management, customer experience and satisfaction, user profiling, recommendation systems, marketing, and brand management. Figure 1 presents the taxonomy of NLP applications in online customer reviews. The taxonomy presented in this survey is critical for understanding the applications of NLP in the analysis of online customer reviews.

### Sentiment analysis and opinion mining

Sentiment analysis involves extracting and classifying opinions, emotions, and attitudes expressed in textual data to determine whether a text expresses a positive, negative, or neutral sentiment. In contrast, opinion mining, also known as subjectivity analysis, focuses on identifying and analyzing subjective information from various sources, such as reviews, social media, and discussions. Many studies have examined deep neural networks and traditional algorithms for sentiment extraction in multiple domains. A study used Bing Liu's aspect-based opinion-mining approach to tourism, addressing specific elements absent from the physical product model. The authors detected elements in online travel reviews and proposed advanced NLP-based sentiment categorization methods. They used a general tool architecture for Lake District TripAdvisor reviews and increased the performance, reaching a 92% F-measure for the sentiment classification (*Marrese-Taylor, Velásquez & Bravo-Marquez, 2014*). A study categorized sentiments and varied emotions from a large dataset of internet mobile phone reviews using sentiment analysis. This comprehensive strategy helped consumers and producers make informed decisions by

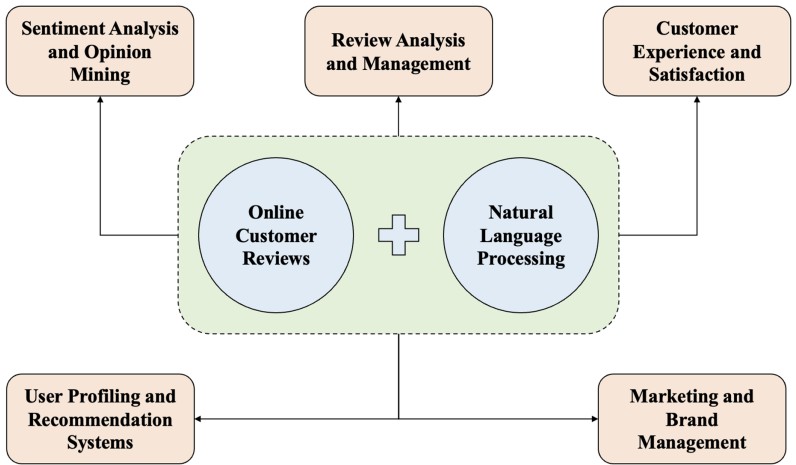

**Figure 1 Taxonomy of applications of NLP in online customer reviews.**

emphasizing the relevance of online reviews in determining client needs and providing timely feedback (*Singla, Randhawa & Jain, 2017*).

A study also used cognitive computing-based Artificial Intelligence (AI) technologies to examine textual content and numerical ratings in online reviews. The study examined hotel reviews using sentiment analysis to uncover discrepancies between review content and scores (*Fazzolari et al., 2017*). Additionally, a study developed a Chinese sentiment mining method that outperformed other models on TripAdvisor reviews and included features to improve sentiment analysis (*Li & Yang, 2017*). A study also used Amazon and Twitter data to develop an NLP and machine learning book review sentiment categorization system. Users might analyze public opinion and create user-friendly word clouds based on top attributes (*Magesh & Swarnalatha, 2017*). Genetic Algorithms for automated text sentiment analysis performed well on huge Amazon datasets and were highlighted for business and scientific applications (*Dufourq & Bassett, 2017*).

Deep learning has been utilized in sentiment extraction using convolutional neural network (CNN) and long short term memory (LSTM) architectures to extract features from customer reviews (*Jain, Kumar & Mahanti, 2018*). Another study examined the role of opinion mining in e-commerce, using algorithms such as naïve Bayes, Support Vector Machine (SVM), random forest, and hybrid SVM to classify reviews as positive or negative, improving understanding and the use of opinion mining in online reviews (*Gawade & Parthiban, 2018*). NLP was also used for Amazon reviews to enhance service by comparing K-nearest neighbour, SVM, and decision tree classification algorithms to analyze customer feedback (*Aryo Prakoso et al., 2018*). Deep learning techniques like word2vec for word embedding and CNN were used to evaluate social marketing tactics and help consumers make informed purchase decisions (*Panthati et al., 2018*). Another study used classical machine learning and deep learning to classify multiple affective attributes with over 90% accuracy using customer emotional needs from online product reviews (*Wang et al., 2018*). A study modified the Opinion Mining system evaluation by using a user profiling system

to parameterize the system based on user preferences and improve results (*Angioni et al., 2018*).

A study examined opinion mining and sentiment analysis of Amazon product reviews to increase accuracy. The proposed Senti algorithm outperformed sentiment analysis APIs, allowing for improved commercial, political, and financial decision-making (*Karthikayini & Srinath, 2018*). Another research used Hybrid Attribute Based Sentiment Classification (HABSC) to identify sentiment orientation in online consumer reviews. HABSC outperformed state-of-the-art approaches by integrating syntactic characteristics, implicit word relations, and domain-specific information to reveal differences in review content and ratings between local and international consumers on multinational social commerce platforms (*Bansal & Srivastava, 2019*). Japanese restaurant reviews were also examined to see how ethnic culture affects customer ratings. Bilingual text mining technique highlighted cultural implications in social commerce by showing different emotion distribution patterns among Japanese and Western customers (*Nakayama & Wan, 2019*).

Researchers also examined sentiment analysis in NLP to evaluate emerging technologies with many aspects. Mobile reviews were identified using web scraping and machine learning techniques, specifically decision tree and SVM, with over 90% accuracy using 1–2 grams. User experience and marketing tactics improved by real-time product review sentiment analysis (*Chauhan & Sehgal, 2019*). A study examined the importance of sentiment analysis in business analytics for product and market competitiveness. The research examined machine learning classification methods, including a hybrid algorithm, emphasizing the rising relevance of sentiment analysis in corporate strategy and product quality (*Jabbar et al., 2019*; *Pathuri et al., 2019*). A study addressed the constraints of using customer ratings or review summaries to extract valuable data from online product reviews (*Dharaiya et al., 2020*). Two new corpora with full word clouds were produced using a general approach and a specific approach to improve product analysis accuracy and efficiency. The approach examined varied consumer sentiment and product characteristics. Additionally, a new sentiment analysis algorithm improved the Dempster-Shafer algorithm (*Jamadi Khiabani, Basiri & Rastegari, 2020*). This novel method treated reviews as sentences with sentiment orientations and ratings. The method outperformed the original algorithm on TripAdvisor and CitySearch datasets.

Lexicon-based analysis was used to evaluate Amazon books and writers (*Khatun et al., 2020*). The study used a bag-of-words technique to evaluate review positivity and negativity, emphasizing the importance of sentiment in market analysis and its ability to anticipate business trends. Another study compared LSTM, random forest, SVM, and eXtreme Gradient Boosting (XGBoost) for sentiment analysis (*Chatterjee et al., 2021a*). The findings highlighted applications in customer management systems and Twitter and e-commerce platforms. Two aspect-based sentiment analysis extraction algorithms were presented to analyze unstructured social media reviews (*Bhamare & Prabhu, 2021*). Using SemEval, Yelp, and Kaggle datasets, the hybrid technique predicted aspect categories accurately. An end-to-end sentiment analysis technique for negotiations was introduced by *Mukherjee et al. (2021)*. The method reduced biases and enhanced sentiment categorization across datasets. Sentiment analysis improved user experience on e-

commerce platforms(*Gupta, Rastogi & Katal, 2021*). Compared to logistic regression, multinomial naïve Bayes, and SVM, stochastic gradient descent has the highest accuracy. A lexicon-based technique and logistic regression were used to analyze Web sentiment (*Khanam & Sharma, 2021*). These methods successfully extracted sentiments from various web sources.

A unique Statistics-Based Outlier Detection and Correction Method study (*Chatterjee et al., 2021b*) highlighted the need for proper sentiment analysis in Amazon customer reviews. This technique improved sentiment analysis without data loss over previous systems. Sentiment analysis was performed on Amazon electronics product reviews using different machine learning algorithms (*Urkude et al., 2021*). Logistic regression had the best accuracy, demonstrating the relevance of sentiment analysis in customer recommendations. Another work used part-of-speech-based feature extraction and game-theoretic rough sets to reduce dimensionality in sentiment analysis (*Chen & Yao, 2021*). The model outperformed other models and classifiers. A study on Amazon Electronics product reviews using machine learning due to the rising relevance of e-commerce (*Hawlader et al., 2021*). Preprocessing methods were tested, and the multi-layer perceptron classifier performed well. A publication introduced the BERT Base Uncased model to improve e-commerce platform review sentiment analysis, outperforming standard machine learning approaches (*Geetha & Karthika Renuka, 2021*). A work using NLP for sentiment analysis addressed Amazon's growing customer review volume (*Sinnasamy & Sjaif, 2022*). The Term Frequency–Inverse Document Frequency (TF-IDF) approach using unigram and SVM was the most accurate for Amazon product reviews.

An Ensemble Classifier study (*Maurya & Pratap, 2022*) stressed the importance of online reviews in understanding customer perspectives and needs. The Ensemble Classifier outperformed machine learning techniques in consumer feedback analysis. A article used naïve Bayes, random forest, and SVM algorithms to improve the accuracy of sentiment analysis for Amazon product reviews (*Naureen, Siddiqa & Devi, 2022*). Aspect-based BERT models were used for tourist sentiment analysis (*Chu et al., 2022*). The findings helped vendors improve their products and services and provided users with personalized recommendations. A mixed generative-discriminative strategy combining Fisher kernels and hidden Markov models improved textual sentiment analysis (*Nasfi & Bouguila, 2022*). Amazon and IMDb user reviews showed that the method improved sentiment identification compared to existing techniques. The influence of digitalization on e-commerce and information overload was examined using machine learning algorithms on Amazon fine food reviews (*Yarkareddy, Sasikala & Santhanalakshmi, 2022*). The authors attempted to simplify review analysis so customers could quickly and accurately assess product opinions. Another research examined BERT-based sentiment analysis across domains (*Roccabruna, Azzolin & Riccardi, 2022*). The study showed that sentiment analysis should incorporate class label variations from various sources.

A study used a hybrid method for sentiment analysis of Amazon customer reviews using NLP, machine learning, and deep learning (*Juyal, 2022*). A study demonstrated that analyzing sentiment can enhance brand value, advertising, and customer service. SVM and CNN models were used to perform sentiment analysis on an e-commerce platform by

*Uma et al. (2022)*. Both methods were found to be more accurate compared to others. A study investigated the effectiveness of using machine learning for review sentiment analysis (*Kamalesh & Vijayalakshmi, 2022*). A study also attempted to utilize NLP to automate the analysis of product reviews from various platforms, such as Amazon (*Singh et al., 2022*). The approach used machine learning to train a neural network to classify product reviews as positive, neutral, or negative. A study examined how varied NLP algorithms affected Yelp and Zappos data (*Yang, Pang & Kan, 2022*). Neural networks and BERT offered insights into algorithm selection that were useful for consumer review analysis. The research used NLP methods like TF-IDF Vectorizer and Count Vectorizer to create a food industry model. Logistic regression, dummy classifier, and random forest classifier were used to efficiently analyze online review sentiments, giving manufacturers significant product perception insights. The investigation found that the proposed sentiment analysis model worked (*Gupta et al., 2022*). Another article proposed a Bayesian network architecture for sentence-level sentiment analysis of e-commerce product reviews with automated rule creation and progressive retainability. The study met model requirements instantaneously, demonstrating its scalability and durability in opinion mining across various domains (*Hariharan et al., 2023*).

A new method using agglomerative clustering for outlier detection and a stacked autoencoder with ensemble classification algorithms was developed to detect sarcastic tweets and reviews. This technique outperformed other algorithms in sarcasm prediction and sentiment identification with 99.3% accuracy (*Maheswari & Dhenakaran, 2023*). With 1.5 million Amazon and Yelp reviews, a study introduced the 'Amazon and Yelp Reviews' dataset for sentiment analysis. The sentiment analysis approach achieved 87.3% accuracy by utilizing a Bidirectional LSTM (BiLSTM) model, user comment preparation, and daily data collection. The dataset and methodology can be used for consumer feedback analysis and online reputation management (*Geetha et al., 2023*). Amazon values customer opinions and stresses the importance of customer satisfaction in organizational success. The article used NLP to turn text into numerical arrays for machine learning techniques. Five scores were assigned to Amazon reviews using supervised machine learning algorithms, such as SVM, naïve Bayes, and decision tree (*Dharrao et al., 2023*). Another article used machine learning to assess Amazon product review sentiment across categories. Text Blob, logistic regression, SVM, and multinomial naïve Bayes improved sentiment classification accuracy, proving that various review sentiment ratings can be predicted (*Kumar, 2023*). A study (*Mehul et al., 2023*) focused on sentiment polarity analysis for e-commerce customer reviews, while (*Solairaj, Sugitha & Kavitha, 2023*) presented an EESNN-SA-OPR method utilizing Collaborative Filtering (CF) and product-to-product similarity.

A study investigated business strategies for customer retention and attraction, employing an NLP-based sentiment analysis (*Sumathi & Santharam, 2023*). The impact of internet reviews on consumer decisions was examined using a CNN model for text review sentiment classification. Comparative analysis showed that the CNN model achieved 90% accuracy using Amazon reviews. Stop words are crucial to sentiment analysis, and the CNN model outperforms other algorithms on big datasets (*Qorich & El Ouazzani, 2023*).

A publication also introduced the Adaptive Particle Grey Wolf Optimizer with Deep Learning Based Sentiment Analysis (APGWO-DLSA) approach for sentiment analysis on online product reviews using NLP and machine learning algorithms. The proposed technique on the Cell Phones And Accessories (CPAA) dataset performed better, obtaining 94.77% accuracy (*Elangovan & Subedha, 2023*). NLP and LSTM were used to create a customer review summary model to handle the increase in textual data. The hybrid sentiment analysis method provided organizations with important insights due to its excellent accuracy (94.46%), recall (91.63%), and F1-score (92.81%) (*Kaur & Sharma, 2023*).

A study on mobile phone reviews utilized consumer reviews to improve post-purchase products. After testing SVM, naïve Bayes, and logistic regression algorithms, the Random Forest (Unigram) classifier performed best on a balanced dataset, highlighting the importance of sentiment analysis in consumer feedback for product development (*Venkataraman & Jadhav, 2023*). LSTM and naïve Bayes were compared for sentiment analysis of online product reviews. Comprehensive assessments of varied internet items were conducted to understand user attitudes better (*Meghana et al., 2023*). A recent article uses data mining to analyze sentiments on Facebook, Instagram, Twitter, and Amazon. The research used consumer input to improve corporate strategy and predict customer requirements. Twitter data obtained *via* the Twitter API was analyzed using NLP techniques, demonstrating their ability to provide organizations with essential insights for personalized marketing and organizational benefit (*Kshirsagar et al., 2023*). The summary of existing literature on sentiment analysis and opinion mining is given in Table S2.

### Review analysis and management

Review analysis and management include studies on analyzing online customer reviews to ensure their authenticity, trustworthiness, quality, and usefulness in e-commerce. Initiatives have been taken to address fake reviews and counterfeit goods in the context of online marketplaces. AI methods like NLP and topic analysis were used to detect counterfeit items on Amazon and eBay. Topic analysis of product and seller reviews identified deception-related keywords and concepts. The findings showed that automated counterfeit detection might boost online marketplace trust and efficiency (*Wimmer & Yoon, 2015*). The fake Product Review Monitoring and Removal System (FaRMS) analyzed reviews from numerous platforms with 87% accuracy in English and Unique Urdu support to combat fake reviews. By providing honest product ratings, FaRMS sought to improve customer satisfaction (*Ata Ur et al., 2019*).

A study examined how review length affects online purchasing decisions and questioned the idea that lengthier reviews are always better. Using Amazon reviews and advanced NLP, the study discovered that argumentation frequency altered the association between review length and helpfulness, showing that longer reviews were not always more helpful (*Lutz, Pröllochs & Neumann, 2019*). A novel approach combined business data and user reviews to improve relevance and diversity in machine-generated fake reviews. The proposed model generated high-quality and diverse reviews in response to seller descriptions (*Jin, Zhang & Zhang, 2019*). The significance of vigilance when dealing with

manipulation on large online platforms was highlighted by stylometry-based algorithms that detected misleading online reviews (*Sadman et al., 2020*).

A predictive model used BERT and deep learning to improve online product review usefulness evaluation and overcome previous model limitations (*Xu, Barbosa & Hong, 2020*). Introducing Social Network Strength (SNS) elements to analyze the influence of friends and followers on review helpfulness enabled overcoming information overload in online customer reviews. The methodology evaluated using Yelp business reviews gave insights to researchers, businesses, reviewers, and review platforms (*Bilal et al., 2021*). In order to combat the ubiquity of fake reviews, supervised machine learning was utilized to identify opinion spammers, which improved the accuracy of spotting fraudulent reviews on well-known platforms (*Jain, Pamula, Ansari 2021*).

Addressing the critical issue of fake review detection, NLP techniques and machine learning models, including Naïve Bayes and random forest, were applied to combat the increasing prevalence of fake reviews in the E-commerce industry. The models demonstrated scalability, offering a solution for platforms to identify and address spam reviews promptly (*Anas & Kumari, 2021*). Another study aimed to determine the most effective feature combination for detecting fake reviews, highlighting the significance of behavior-related features in combination with text-related features, with verified purchases emerging as a crucial factor (*Birim et al., 2022*). A hybrid CNN-LSTM deep learning model with sentiment analysis techniques was employed to assess the authenticity of customer reviews, proposing a solution to combat fraudulent reviews in the e-commerce sector (*Aishwarya & Prashanth Kumar, 2023*).

Supervised machine learning and NLP techniques were utilized to identify and remove fake reviews from a dataset, focusing on significant E-commerce platforms to combat the prevalence of counterfeit product reviews impacting customer decisions and profits (*Thilagavathy et al., 2023*). A Python-based system was introduced to detect fake product reviews on Amazon, using SVM techniques to distinguish between genuine and fake reviews and enhance the reliability of product reviews (*Iliev et al., 2023*). Lastly, an innovative method employing a CNN and adaptive particle swarm optimization with NLP techniques achieved a remarkable 99.4% accuracy in identifying fake online reviews, offering practical implications for consumers, manufacturers, and sellers in maintaining the trustworthiness of online reviews (*Deshai & Bhaskara Rao, 2023*). Another study proposed a generalized solution by fine-tuning the BERT model to predict review helpfulness, demonstrating better performance compared to traditional bag-of-words methods (*Bilal & Almazroi, 2023*). The summary of existing literature on review analysis and management is given in Table S3.

### Customer experience and satisfaction

The term customer experience includes all customer interactions with a company, from browsing online to receiving support after purchase. These interactions eventually affect the customer's perception and loyalty towards the company. On the other hand, satisfaction refers to the degree of contentment that a customer feels with the products or services received. This can often be determined through customer feedback, reviews, and

ratings. A study adopted Bing Liu's aspect-based technique to identify customer preferences in TripAdvisor hotel and restaurant reviews to examine opinion mining in tourism. The approach demonstrated 90% precision and recall in extracting sentiment orientations, though it struggled with explicit aspect expressions. Emphasizing the value of tourism product reviews, the research underscored the importance of aspect-based opinion mining in revealing customer preferences (*Marrese-Taylor et al., 2013*). Another study focused on English online reviews of hotels, employing NLP and sentiment analysis. Organizations emphasizing these techniques outperformed peers in growth, earnings, and performance metrics, offering practical implications for hotel managers to leverage social media reviews for strategic decision-making (*Tian et al., 2016*).

Introducing a novel method for hotel summaries from travel forums, a study incorporated author credibility and conflicting opinions. Using a new sentence importance metric and k-medoids clustering algorithm, the approach outperformed conventional methods, confirmed by subjects for providing more comprehensive hotel information (*Hu, Chen & Chou, 2017*). In retail, an article proposed an online platform using NLP to analyze customer sentiments and streamline input through speech-to-text technology. The focus was on enhancing the shopping experience by understanding emotions expressed in reviews and suggesting smart shop solutions to improve overall customer satisfaction (*Rogojanu et al., 2018*). Another research attempted to create artificial personal shoppers for e-commerce platforms, emphasizing user engagement and trust. The study adapted existing information retrieval and NLP technologies to establish the groundwork for effective voice-based artificial personal shoppers in the online shopping domain (*Limsopatham, Rokhlenko & Carmel, 2018*).

A unique approach assessed customer loyalty through sentiment analysis of online reviews, achieving a 94% accuracy in determining loyalty types. Leveraging tokenization, lemmatization, and SentiWordNet, the study utilized a fuzzy logic model with rule-based systems, surpassing previous methods (*Ghani, Bajwa & Ashfaq, 2018*). Another study used NLP to extract insights from user-generated product reviews to address the need for general e-commerce platforms for vitamins and nutraceuticals. The system provided a five-point rating system, summarized commonly discussed topics, and offered representative reviews, empowering consumers with tailored information for informed decisions (*John et al., 2019*). A study proposed a rapid customer loyalty model for e-commerce with a 72% loyalty rate from Amazon.com reviews (*Ashfaq & Kausar, 2019*). Similarly, sentiment analysis and opinion mining in Yelp datasets using aspect-based sentiment analysis provided business strategies based on 1-year forecasted data, emphasizing the importance of leveraging online reviews to improve customer satisfaction (*Ching & De Dios Bulos, 2019*).

Analyzing user-generated hotel review data comprehensively, a study employed various techniques, achieving high precision (0.95) and recall (0.96). Visual analytics revealed patterns in user ratings, emphasizing the potential for improving business services and product quality (*Chang, Ku & Chen, 2019*). Investigating the impact of latent content factors on online review helpfulness, the study found that argument quality and valence significantly influenced review helpfulness. This approach surpassed previous manifest

content and reviewer-related factors, enhancing understanding and addressing sentiments for improved customer satisfaction (*Srivastava & Kalro, 2019*). A study reviewed online reviews of hotel guest experiences and distinguished patterns between Asian and non-Asian guests, revealing service failures in different domains and stages of the hotel guest cycle (*Sann & Lai, 2020*).

Leveraging logistic regression and NLP, another study identified sentiment and topics among tourists in Cyprus, offering insights into the complex relationship between tourist culture, purchasing power, and reviews (*Christodoulou, Gregoriades & Papapanayides, 2020*). A study addressed the problem of processing user feedback quickly and effectively by introducing a crowdsourcing technique for categorizing app store reviews, suggesting that crowdsourcing has the potential to be a reliable and low-cost source for classifying user reviews (*van Vliet et al., 2020*). Investigating parental preferences for childcare using Yelp reviews revealed variations in satisfaction based on income levels, emphasizing safety, learning environment quality, and child-teacher interactions as pivotal factors (*Herbst et al., 2020*). Exploring opinion summarization in Web 3.0 platforms, a study proposed a novel graph-based abstractive technique, comparing it with extractive methods for coherence and completeness in generating summaries (*Bhatia, 2021*).

Enhancing review-based question-answering systems using NLP models, a study addressed the challenge of manual handling of product-related queries on online platforms. The proposed enhancements, including BERT, significantly improved response effectiveness, achieving a BLEU score of 0.58 (*Moharkar et al., 2022*). Analyzing visitor reviews of Croatia's Plitvice Lakes National Park, a study utilized multidimensional scaling, sentiment analysis, and NLP to identify key topics and analyze strengths and weaknesses, providing valuable insights for protected natural areas (*Sergiacomi et al., 2022*). A study identified critical elements influencing success through text mining and sentiment analysis on TripAdvisor reviews, highlighting the significance of tour guides in consumer satisfaction (*Barbierato, Bernetti & Capecchi, 2022*). Furthermore, a study introduced a hierarchical attention network-based framework for analyzing Amazon Smartphone reviews (*Ratmele & Thakur, 2022*).

A study used sentiment analysis to classify smartphone reviews and predict product ratings based on user feedback (*Suresh & Gurumoorthy, 2022*). Addressing information overload in Community-based Question Answering (CQA) platforms, a study introduced a CQA summarization task. Evaluating various summarization methods, the research provided a robust baseline for CQA summarization, contributing to the user experience in navigating overwhelming information (*Hsu, Suhara & Wang, 2022*). Assessing pre-trained transformers for sentiment extraction, a study applied five models to an Amazon database of automotive products, suggesting their potential for practical applications like product monitoring and market research (*Alves, Lobo & Reis, 2022*). Lastly, employing machine learning and NLP, a study demonstrated the effectiveness of text summarization in efficiently handling and comprehending extensive online product review data (*Nainwal et al., 2023*). The summary of existing literature on customer feedback and satisfaction is given in Table S4.

### User profiling and recommendation systems

User profiling and recommendation systems aim to improve personalized user experiences across different platforms. Two innovative algorithms were introduced to address challenges related to service recommendation accuracy and incomplete modalities in recommender systems. Value Features and Distributions for Accurate Service Recommendation (VFDSR) leverages fine-grained value features extracted from customer reviews to enhance personalized service recommendations, demonstrating better performance on a Yelp dataset (*Wang et al., 2017*). Learning to recommend with missing modalities (LRMM) tackles incomplete modalities through modality dropout and a multimodal sequential autoencoder, outperforming existing methods and proving robust in mitigating data sparsity and the cold-start problem (*Wang, Niepert & Li, 2018*). Integrating data mining, human psychology, and NLP aimed to enhance recommender-based mobile applications. The strategy generated "wh" questions from recommended items, utilizing a web scraper for relevant information, and strategically employed human-computer interaction psychology to increase user engagement. Survey results confirmed an improved hit rate, supporting the method's effectiveness on platforms like Amazon (*Neeraj, Oswald & Sivaselvan, 2018*).

Another study focused on the rising use of intelligent personal assistants in business workflows, introducing an explanation mode feature for speech interaction in Enterprise Resource Planning software. Task attraction was identified as pivotal for usefulness, emphasizing the supplementary role of intelligent personal assistants alongside traditional input methods (*Hüsson, Holland & Sánchez, 2020*). Advancements in personalized advertising and recommender systems were explored with Double Attention for Click-Through Rate Prediction (DAMIN), an enhanced model incorporating a double attention mechanism into the Deep Interest Network. Experimental results on Amazon datasets demonstrated DAMIN's effectiveness, improving AUC by 4–5% compared to classical models (*Xia, Fang & Shi, 2021*). TripAdvisor data was leveraged to enhance hotel customer targeting through a fine-tuned BERT model and a multi-criteria recommender system. Outperforming a benchmark single-criteria system, the approach considered varied hotel aspects, demonstrating better performance (*Zhuang & Kim, 2021*). In the Pakistani fashion industry, user interests were extracted from social media using Latent Dirichlet Allocation (LDA), Latent Semantic Analysis (LSA), and BERT for topic modeling, sentiment analysis tools, and K-Means for clustering. Empirical validation demonstrated moderate agreement between human and machine evaluations (*Tahir & Asif Naeem, 2022*).

An innovative product recommender model for e-commerce platforms analyzed customer reviews using NLP, sentiment analysis, and clustering algorithms. Experiments on Amazon datasets showed notable enhancement in multi-node cluster setups over single-node configurations (*Patidar & Patel, 2022*). A graph-based movie recommender system incorporating user sentiments and emotions demonstrated better performance. Utilizing BERT for sentiment analysis and a Kaggle dataset, the proposed IGMC-based models outperformed conventional and state-of-the-art graph-based systems (*Lee et al., 2022*). The impact of cognitive absorption dimensions on continuous use intention in AI-
driven Recommender Systems was investigated, revealing that curiosity and focused immersion significantly influenced continuous use intention (*Acharya, Sassenberg & Soar, 2024*).

The study proposed a novel approach to enhance tourist trip suggestions by integrating neural networks and deep learning techniques. The hybrid framework combined Neural Network-LSTM for Point of Interest recommendations and BERT for sequential trip recommendations, demonstrating better performance on TripAdvisor and Yelp datasets (*Noorian, Harounabadi & Hazratifard, 2023*). A weighted hybrid recommendation method combining user reviews, rating data, and sentiment analysis improved precision scores on the Amazon Reviews dataset, integrating CF for enhanced recommendations (*Paul & Singh, 2023*). FusionSCF addressed issues in Recommendation Systems by integrating CF with sentiment analysis of textual user reviews. Using e-commerce datasets, the model combined weighted ratings and sentiment scores to enhance recommendations, demonstrating the effectiveness of the sentiment-based model over traditional CF methods. The study also explored the impact of fake reviews on the filtering system (*Ananth, Raghuveer & Vasanth Kumar, 2023*). The summary of existing literature on user profiling and recommendation systems is given in Table S5.

### Marketing and brand management

Marketing involves promoting and selling products or services to target audiences, whereas brand management involves managing a brand's reputation, perception, and value in the marketplace. Advanced methods for opinion mining in concise e-commerce feedback were studied by a study to create seller rating profiles. The novel approaches integrated opinion mining and dependency relation analysis to propose an algorithm for extracting dimension ratings. The computation of dimension weights from ratings was framed as a factor analytic problem and solved through matrix factorization. The algorithm demonstrated its effectiveness on eBay and Amazon datasets, achieving 93.1% and 89.64% accuracy in identifying dimensions and ratings, respectively (*Cui et al., 2013*). CommTrust, a novel approach to the 'all good reputation' problem in e-commerce trust models, leveraged free-text feedback comments to create a multidimensional trust model. The algorithm, combining NLP, opinion mining, and topic modeling, effectively mitigated universally high seller reputation scores on eBay and Amazon, providing a more reliable ranking of sellers based on trust (*Zhang, Cui & Wang, 2014*).

Another study combined opinion mining and CF algorithms to analyze Yelp data, highlighting inconsistencies between textual reviews and star ratings. The research explored the impact of restaurant popularity on user ratings, yielding improved results with a lower root mean squared error (RMSE) (*Angioni, Clemente & Tuveri, 2015*). The significance of consumer reviews in e-commerce was emphasized in a distinct model that focused on fine-grained analysis of feedback. The methodology, validated on Amazon and Flipkart, revealed notable discrepancies in trust scores, enhancing the understanding of seller trust profiles (*Bhargava et al., 2016*). The impact of Amazon's Verified Purchase badge on review helpfulness and product ratings was investigated, revealing significant

increases in review helpfulness and product ratings for verified purchase reviews (*Kokkodis & Lappas, 2016*).

A study used various relevance algorithms to enhance Amazon Search's relevance ranking, emphasizing the significant impact on customer satisfaction and financial outcomes (*Sorokina & Cantú-Paz, 2016*). Different methods for analyzing consumer opinions on platforms like Amazon were explored, introducing a hybrid approach that effectively ranked products based on text reviews, question answer (QA) data, and star ratings, enhancing sales predictions (*Anjum & Sabharwal, 2016*). The study addressed the financial and reputational impact of product issues in Over-the-Counter (OTC) pain relief products, utilizing Amazon's product reviews to identify safety and efficacy concerns through "smoke word" dictionaries and sentiment analysis (*Adams, Gruss & Abrahams, 2017*). Another research investigated whether models trained on a dataset could accurately reflect human proficiency in online review writing, employing knowledge tracing to track the development of reviewers' skills over time (*Megasari et al., 2018*).

The challenge of navigating through lengthy customer reviews was addressed through a multi-criteria decision-making approach to recommend optimal products on platforms like Flipkart and Amazon (*Kumar, 2018*). The detection of ironic opinions in social networks and e-commerce was explored, and feature-based irony detection was compared with a novel approach using character language model classifiers, showing competitive accuracy in experiments (*Clavel Quintero & Arco García, 2018*). The evolving landscape of consumer behavior in e-commerce was examined, and an algorithmic solution was proposed to mitigate inaccuracies in user-generated reviews and enhance the decision-making process using NLP techniques (*Mastan Rao et al., 2018*). The Ranking Hotels using Aspect Level Sentiment Analysis (RHALSA) algorithm was introduced, effectively evaluating and ranking hotels based on user reviews through aspect-level sentiment analysis on a Tripadvisor dataset (*Panigrahi & Asha, 2018*).

Leveraging user-generated content for marketing was explored through sentiment analysis tools, proposing a framework to derive new scores reflecting consumer sentiments for distinct product features on Amazon (*Kauffmann et al., 2019*). The impact of technology on people's lifestyles and decision-making processes was investigated using Yelp, emphasizing the importance of reviews analysis in monitoring changes in business public opinion over time (*Galli et al., 2019*). A Feature-Based Product (FBP) Recommendation system using NLP and sentiment analysis on Amazon mobile product reviews was proposed, demonstrating the effectiveness of SVM in suggesting the best company products for user-requested features (*Koneru et al., 2019*). The Level of Success model (LOS) was introduced, employing NLP, review quantification, and image analysis to contribute valuable insights for effective product market evaluation in the Amazon online market review context (*Li et al., 2020*).

Quantifying online brand image (OBIM) by analyzing consumer reviews was explored, introducing a model that evaluated associations' favorability, strength, and uniqueness through sentiment and co-word network analysis (*Mitra & Jenamani, 2020*). Using Python for preprocessing NLP features, the study focused on product recommendations on Amazon, revealing insights for quarterly sales forecasting and product development trends

based on customer reviews (*Yang et al., 2020*). A novel method, Tagging Product Review (TPR), was introduced to summarize e-commerce product reviews, achieving high tag relevance scores for both popular and cold products on Amazon (*Konjengbam, Kumar & Singh, 2020*). AmazonRep, a reputation system that considers review sentiment, helpfulness votes, review timing, and user credibility, proved effective in generating and presenting reputations for diverse products on Amazon (*Benlahbib & Nfaoui, 2020b*). Reputation generation for diverse entities using customer reviews was addressed through a unified reputation value integrating helpfulness, time, rating, and sentiment. The method outperformed three existing systems, offering a comprehensive approach and visualizations for numerical reputation, opinion categories, and top reviews (*Benlahbib & Nfaoui, 2020a*).

A study validated sustainable features for French Press coffee carafes extracted from Amazon reviews using a novel design technique called collage placement. The study revealed a disparity between customer perceptions and engineered sustainability, emphasizing the importance of understanding diverse perspectives. Participants evaluated products based on social, environmental, and economic sustainability, highlighting the efficacy of the collage method in assessing sustainability perceptions. Demographic variations in sustainability perceptions further underscored the method's relevance (*El-Dehaibi, Liao & MacDonald, 2021*). Another research focused on online sales strategies for Amazon products, using sentiment analysis and opinion mining for microwave ovens, baby pacifiers, and hairdryers. Mathematical models evaluated product reputation trends, predicting potential success or failure and proposing design features for enhanced desirability (*Liu, 2021*).

The study on online product reviews from Flipkart and Amazon employed sentiment analysis and a bag-of-words model to assess the impact on third-party sellers. Categorizing reviews and conducting topic modeling, the findings emphasized the importance of considering both product and seller reviews for a seamless delivery and defect-free product that benefits both buyers and sellers (*Nellutla et al., 2021*). Introducing a novel approach for computing reputation scores, the article utilized a BiLSTM, recurrent neural network (RNN), and NLP techniques to analyze textual opinions on online platforms like IMDB and Amazon. Experimental results demonstrated the method's effectiveness, aligning closely with ground truth and suggesting practical applicability for reputation generation (*Boumhidi, Benlahbib & Nfaoui, 2021*).

A study on TripAdvisor reviews and online weather data used NLP to assess the impact of weather conditions on tourists' intention to revisit a destination. Enriching the dataset with weather information and hotel ratings, the findings identified factors like heat index and weather disparities influencing revisit intention, providing valuable insights for destination managers (*Christodoulou et al., 2021*). Twitter content was evaluated in a unique reputation generation system to determine credible reputation scores for products. Integrating sentiment orientation, user credibility, and tweet credibility, the system's computed values are closely aligned with ground truth scores from various platforms. This suggests practical applications for consumers and businesses in decision-making processes on e-commerce platforms (*Boumhidi & Nfaoui, 2021*).
Analyzing Amazon and iHerb reviews, the research on sweetness in food products identified opportunities for less sweet products catering to a healthier consumer base. The study used manual curation, NLP, and machine learning to reveal the impact of sweetness on product liking, suggesting potential benefits for health-conscious customers and manufacturers (*Asseo & Niv, 2022*). Challenging the belief that longer product reviews are uniformly more helpful, the study utilized advanced machine learning methods to analyze Amazon reviews' sentence-level argumentation. Contrary to prevailing views, longer reviews with frequent shifts between positive and negative arguments were perceived as less helpful, with implications for optimizing customer feedback systems and improving reviewer guidelines (*Lutz, Pröllochs & Neumann, 2022*).

The analysis of customer reviews for small domestic robots on Amazon addressed failure types and their impact on customer experience. Technical failures, particularly related to task completion and robustness, significantly impacted customer experience more than interaction or service failures. An NLP model predicted failure content in reviews, providing insights for prioritizing crucial issues for robotic system improvement (*Honig et al., 2022*). The study explored the memorable tourist experience (MTE) concept using TripAdvisor reviews, employing NLP and machine learning to analyze terms and relationships. Comparative analysis of UNESCO sites revealed shared MTE elements and validated hypotheses, emphasizing the value of reviews as supplementary data in tourism studies (*Sánchez, 2022*).

The impact of digitization on e-commerce was investigated through sentiment analysis of Amazon customer product reviews, utilizing SVM and deep learning techniques. The study provided valuable insights for businesses in the dynamic e-commerce market, indicating the effectiveness of both SVM and deep learning approaches in discerning sentiments (*Arora, Srivastava & Ananda Kumar, 2022*). Introducing the NLP-AHP method, the research assessed online shopping platform reviews through an empirical examination of microwave oven reviews on Amazon. The method swiftly identified crucial comments and temporal patterns, offering a valuable tool for data-driven decision-making to enhance product quality and refine sales strategies (*Tang & Guo, 2022*).

The analysis of Banglish text on social media in Bangladesh employed NLP techniques and machine learning models for product market demand assessment. Results indicated high accuracy in demand analysis, providing valuable insights into popular smartphone choices by gender in the Bangladeshi market (*Hossain, Nayla & Rassel, 2022*). The study on managerial responses to online customer complaints and negative reviews integrated justice theory and service recovery literature. Positive managerial responses influenced future review valence, with rational cues to procedural unfairness complaints enhancing future valence. The article provided insights for both theory and practical applications (*Ravichandran & Deng, 2023*).

An NLP analysis of Amazon reviews explored user satisfaction with physical activity trackers. Sentiment analysis and a Transformer-based language model classified technical aspects and user sentiments, revealing hidden perspectives on product satisfaction (*Mantilla-Saltos et al., 2023*). The study on TripAdvisor used deep learning models based on the Myers-Briggs Type Indicator (MBTI) to discern consumers' personalities from

electronic word-of-mouth (e-WOM). Findings linked specific discussion themes to personality traits, offering insights for personalized marketing messages and optimizing communication strategies (*Christodoulou & Gregoriades, 2023*).

The research addressed the challenge of assessing product quality in e-commerce, introducing the QLeBERT approach. Combining a quality-related lexicon, N-grams, BERT, and BiLSTM for classification, QLeBERT achieved better performance, providing a deeper understanding of textual input for predicting product quality (*Ullah et al., 2023*). An algorithm was introduced utilizing language-transformer technologies for automated product requirement generation from E-Shop reviews. The study showcased the transformative potential of transformer-enhanced opportunity mining in requirements engineering, efficiently extracting critical user needs from consumer reviews to enhance product improvement (*Harth et al., 2023*).

The impact of the "Amazon effect" on consumer perceptions of service attributes in offline/online retailers was explored. Analyzing social media comments using NLP, the study identified triggers for the Amazon effect, highlighting widespread dissatisfaction and reduced satisfaction with other retailers influenced by elevated consumer expectations shaped by Amazon (*Vollero, Sardanelli & Siano, 2023*). The study on CF recommendation systems utilized sentiment analysis on user reviews to derive implicit ratings, introducing novel approaches that demonstrated effectiveness in enhancing CF performance (*Al-Ghuribi, Noah & Mohammed, 2023*).

To address issues in review-based recommender systems, the article introduced the Time-Varying Attention with a Dual-Optimizer (TADO) model, combining a dual-optimizer network, BERT, and time-varying feature extraction. Tested on Amazon Product Reviews datasets, TADO outperformed state-of-the-art techniques by significant margins, offering improved classification and regression losses for enhanced performance (*Li et al., 2023*). Focusing on categorizing customer reviews on Amazon, the study employed machine learning techniques to enhance the e-commerce shopping experience. The model predicted sentiment, aiding users in making informed purchasing decisions by categorizing customer reviews based on inherent attributes (*Harsha et al., 2023*). The summary of existing literature on marketing and brand management is given in Table S6.

## DISCUSSION

The NLP applications in e-commerce research presented in Section 'Taxonomy of NLP Applications in Online Customer Reviews' provide important insights into varied data sources and approaches. Figure S2 shows the distribution of selected and reviewed publications from 2013 to 2023. This survey includes 154 publications representing the evolution of NLP research for online consumer reviews. The number of articles published from 2013 to 2023 shows a growing interest in applying NLP in online customer review analysis. Researchers have explored customer reviews analysis applications in several areas as businesses realize the strategic benefit of knowing customers' opinions. The continual increase of articles shows a dedication to tackling emerging challenges, including fake review identification, cross-domain sentiment transfer learning, and multi-modal analysis.

## RQ1: application areas

Sentiment analysis is a major focus of existing NLP research on online consumer reviews. Early sentiment classification models (*Jabbar et al., 2019*; *Magesh & Swarnalatha, 2017*; *Marrese-Taylor, Velásquez & Bravo-Marquez, 2014*) were gradually replaced by advanced techniques such as deep neural networks (*Geetha & Karthika Renuka, 2021*; *Jain, Kumar & Mahanti, 2018*; *Qorich & El Ouazzani, 2023*) and the incorporation of cutting-edge models like BERT (*Geetha & Karthika Renuka, 2021*; *Singh et al., 2022*). This trend highlights the need for more complex and context-aware sentiment analysis to better grasp online reviewer opinions and emotions. The study of sentiment polarity analysis for e-commerce reviews (*Mehul et al., 2023*) and the proposed EESNN-SA-OPR approach (*Solairaj, Sugitha & Kavitha, 2023*) using CF and product-to-product similarity demonstrate the integration of sentiment analysis with user profile and recommendation systems. Review analysis and management are crucial to NLP applications, as shown in research on handling various online user data (*Bhamare & Prabhu, 2021*; *Karthikayini & Srinath, 2018*). Novel aspect-based sentiment analysis methods (*Chu et al., 2022*; *Uma et al., 2022*) and HABSC, which uses grammatical characteristics and domain-specific information (*Bansal & Srivastava, 2019*), demonstrate a rising focus on extracting granular insights from reviews. The algorithm QLeBERT (*Ullah et al., 2023*) predicts product quality using a quality-related vocabulary, emphasizing sentiment analysis and product evaluation and management. These efforts improve review understanding and digital consumer opinion management.

Another important NLP application in this field is customer experience and satisfaction analysis. Using aggregated sentiment ratings and fuzzy logic to estimate customer loyalty (*Ghani, Bajwa & Ashfaq, 2018*) illustrates efforts to quantify consumer happiness using sentiment analysis. The quick customer loyalty model for e-commerce (*Ashfaq & Kausar, 2019*) and the Level of Success model (LOS) (*Li et al., 2020*) also recognize the need to turn sentiment analytics into customer satisfaction and retention efforts. The use of NLP to extract insights from user-generated reviews in nutraceutical retail (*John et al., 2019*) and the study of sweetness in online food product reviews (*Asseo & Niv, 2022*) demonstrate the multifaceted uses of NLP in customer satisfaction. Implementing NLP with recommendation systems shows its importance in user experiences and purchase decisions. FusionSCF (*Ananth, Raghuveer & Vasanth Kumar, 2023*), which combines CF with sentiment analysis, addresses cold-start and long-tail recommendation system problems. The DAMIN model (*Xia, Fang & Shi, 2021*) for click-through rate prediction and BERT models for a multi-criteria hotel recommender system (*Zhuang & Kim, 2021*) demonstrate how recommendation algorithms may be tailored to user preferences and review sentiments. The study of intelligent personal assistants in company processes (*Hüsson, Holland & Sánchez, 2020*) shows how NLP-driven recommendation systems affect user interactions and experiences beyond e-commerce.

NLP applications in online customer review analysis affect marketing and brand management. The 'Amazon effect' on consumer perceptions of service attributes in Italian consumer electronics retailers (*Vollero, Sardanelli & Siano, 2023*) and sentiment analysis

of online sales strategies for various products (*Liu, 2021*) show how sentiment insights affect marketing strategies. Creating an Amazon customer reputation system (*Benlahbib & Nfaoui, 2020b*) and studying a comprehensive approach to reputation generation from customer reviews (*Benlahbib & Nfaoui, 2020a*) emphasize the importance of sentiment analysis in brand image management. NLP is used to analyze smartphone demand using social media data (*Hossain, Nayla & Rassel, 2022*), demonstrating the importance of sentiment analysis in marketing and customer preferences. Online customer review NLP applications demonstrate language processing technology adaptability and influence. NLP is essential for extracting insights from online user-generated content, including sentiment analysis, review management, customer experience and satisfaction, recommendation systems, and marketing and brand management.

The taxonomy of NLP applications in online customer review analysis highlights the interrelationship between various categories. It is seen that sentiment analysis and opinion mining intersect with customer experience and satisfaction. The advancements in sentiment analysis techniques improve understanding of online customer opinions and influence customer satisfaction analysis. The integration of sentiment analysis with user profiles and recommendation systems highlights the interconnectedness of sentiment analysis with broader aspects of consumer behavior analysis, as evidenced by studies like the EESNN-SA-OPR approach. Furthermore, aspect-based sentiment analysis and HABSC intersect with the customer experience and satisfaction analysis. The fusion of sentiment analysis with recommendation systems, as shown by FusionSCF and the DAMIN model, further blurs the boundaries between sentiment analysis and recommendation algorithms. Moreover, insights from sentiment analysis directly impact marketing strategies, as seen in studies analyzing the 'Amazon effect' on consumer perceptions and the sentiment analysis of online sales strategies. Creating reputation systems based on customer reviews highlights the role of sentiment analysis in shaping brand image and reputation management.

## RQ2: data sources

Existing literature uses data from several review platforms. Figure 2 shows that Amazon's enormous customer review dataset is regularly used in studies. The usage of Amazon datasets across several years suggests a persistent interest in online customer reviews analysis in the context of online shopping. This implies that online product reviews influence customer perceptions and decisions. TripAdvisor is famous for being used in research on travel sentiment. Hotel and visitor reviews on the review platforms help to understand customer attitudes. The regularity with which TripAdvisor is mentioned emphasizes its significance as a source of sentiment-rich data in the travel industry. Studies also employ several datasets, indicating a move towards cross-domain sentiment analysis. Combining IMDb reviews with Amazon data or using varied datasets from Yelp, IMDb, and Kaggle shows a holistic approach to sentiment analysis that considers opinions from different areas. Tourism, e-commerce, and social media information show that sentiment research may be applied across industries. Studies like (*Wimmer & Yoon, 2015*) provide an AI framework for counterfeit goods identification, demonstrating sentiment analysis beyond review websites.

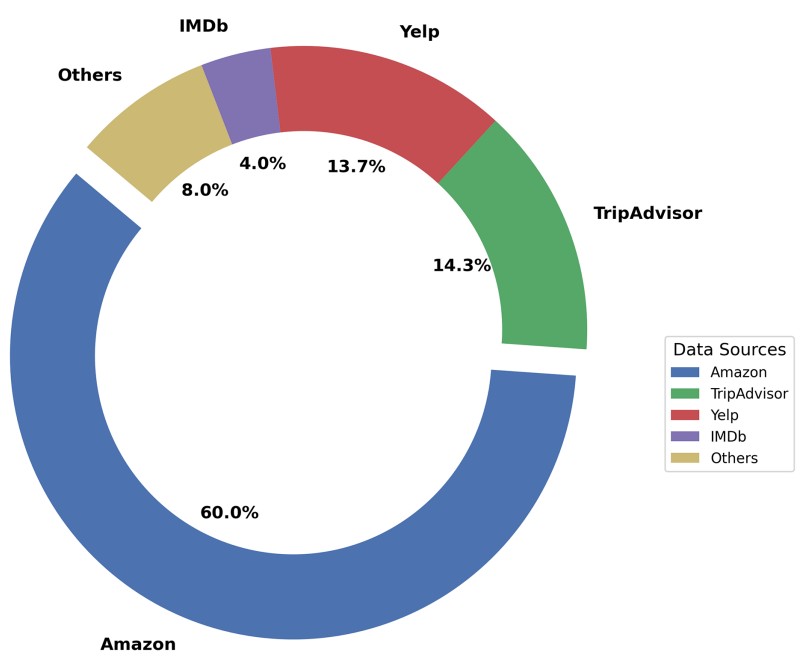

**Figure 2  Distribution of data sources.**

## RQ3: techniques and methods

The changing landscape of techniques has transformed analysis of online customer reviews. SVM and logistic regression are still used, although sophisticated methods are becoming more popular. Deep learning models, such as deep neural networks and LSTM, and cutting-edge transformers like BERT have shown their ability to capture complex textual relationships (*Chatterjee et al., 2021a*; *Geetha & Karthika Renuka, 2021*; *Jabbar et al., 2019*; *Jain, Kumar & Mahanti, 2018*; *Nasfi & Bouguila, 2022*; *Patidar & Patel, 2022*; *Sinnasamy & Sjaif, 2022*; *Zhuang & Kim, 2021*). The use of cutting-edge algorithms like CNN and CF diversifies sentiment analysis (*Elangovan & Subedha, 2023*; *Geetha & Karthika Renuka, 2021*). Ensemble classifiers like random forest (*Kamalesh & Vijayalakshmi, 2022*; *Maurya & Pratap, 2022*) and hybrid systems like CNN-LSTM architectures (*Acharya, Sassenberg & Soar, 2024*; *Juyal, 2022*; *Suresh & Gurumoorthy, 2022*) demonstrate effective integration of several techniques for sentiment analysis. Researchers believe this technological transition is essential from conventional methodologies to advanced integration of cutting-edge technology, notably in NLP and machine learning. Advances in deep learning models and human emotion subtleties reveal ongoing efforts to understand the complexities of sentiment analysis, marking a milestone in technology and emotion comprehension (*Elangovan & Subedha, 2023*; *Jain, Kumar & Mahanti, 2018*; *Sinnasamy & Sjaif, 2022*).

NLP techniques, notably sentiment analysis, are being integrated into the e-commerce sector, with substantial implications for platforms like Amazon, Flipkart, and eBay. Studies show that sentiment analysis may be used to detect fake reviews and understand how cultural variables affect sentiment expression in the e-commerce business (*Arora, Srivastava & Ananda Kumar, 2022*; *Karthikayini & Srinath, 2018*; *Roccabruna, Azzolin &*

*Riccardi, 2022*). CNN-LSTM models, SVM, and hybrid techniques show the dedication to addressing these issues. Sentiment analysis, CF, and novel approaches like FusionSCF and TADO improve recommendation systems by enhancing the user experience and offering personalized information (*Cui et al., 2013*). Sentiment analysis, NLP, and speech recognition are used together to derive insights from user-generated content in e-commerce (*Limsopatham, Rokhlenko & Carmel, 2018*; *Rogojanu et al., 2018*). Recent studies show that sentiment analysis is moving beyond Twitter and Facebook to embrace a wider digital environment, highlighting the increased interest in understanding attitudes across different online platforms (*Gawade & Parthiban, 2018*).

The extent to which sentiment analysis affects e-commerce decision-making is highlighted by its effects on corporate strategy and consumer retention. Researchers study Twitter, YouTube, and Facebook opinions to understand user feelings in the wider online setting (*Sumathi & Santharam, 2023*). Sentiment analysis to identify fake reviews tackles online disinformation and emphasizes the importance of online review reliability. Studies on sentiment analysis in mobile phones, hotels, and the food business show how context affects sentiment. Domain-specific sentiment analysis algorithms are needed to capture and analyze sentiments in varied e-commerce industries (*Barbierato, Bernetti & Capecchi, 2022*; *Sergiacomi et al., 2022*). Thus, sentiment analysis is a strategic instrument that interprets user sentiments and improves business approaches in e-commerce.

In this survey, the number of articles may not reflect the volume of work done in each application area. Keyword searches may exclude articles from particular application areas, limiting their numerical representation. Rather than relying merely on the number of articles, it is preferable to investigate the specific application areas themselves. This will reduce keyword search bias by providing a deeper understanding of each application area in NLP-based review analysis.

### RQ4: open challenges and future directions

The application area of NLP for analyzing online customer reviews has made significant progress, as indicated by the number of articles discussed in Section 'Taxonomy of NLP Applications in Online Customer Reviews'. However, this advancement also introduces several challenges. The challenges and future directions are outlined below to guide future research and facilitate progress in this domain.

*Handling Diverse Data Sources*: For reliable and adaptive textual review analysis algorithms, varied data sources must be managed effectively. The existing research studies emphasize the importance of understanding the complexities of various social media platforms, such as Twitter and Facebook. Opinion mining requires specialized methodologies for different online user data, as shown by *Karthikayini & Srinath (2018)*. The HABSC approach (*Bansal & Srivastava, 2019*) uses syntactic characteristics and domain-specific information to improve sentiment analysis in TripAdvisor and Amazon datasets. Focusing on Amazon electronics product review sentiment research (*Urkude et al., 2021*) highlights the platform's unique issues. With BERT-based models on Twitter, YouTube, Facebook, Amazon, TripAdvisor, Opera, and Personal Healthcare Agent, *Roccabruna, Azzolin & Riccardi (2022)* analyses Italian corpora for sentiment. *Kshirsagar*

*et al. (2023)* mined Twitter data for sentiment analysis while (*Mastan Rao et al., 2018*) suggested using NLP to solve erroneous user-generated reviews in e-commerce. Furthermore, *Boumhidi & Nfaoui (2021)* proposed a Twitter reputation generating method that handles multiple sources. These studies demonstrate that sentiment analysis is complex and requires advanced techniques to handle varied data sources. Federated averaging with weighting (*Karthikayini & Srinath, 2018*; *Urkude et al., 2021*) ensures model adaptation and performance in multiple online ecosystems with varied expressions.

*Aspect-based sentiment analysis*: Aspect-based sentiment analysis is used in NLP to analyze and understand opinions expressed in online customer reviews by focusing on specific aspects of products, services, or experiences. It breaks down the review into different aspects and evaluates the sentiment associated with each aspect separately instead of treating the entire review as a single sentiment. Hence, aspect-based sentiment analysis offers essential opportunities for research and enhancement. Despite advances in sentiment analysis (*Nellutla et al., 2021*; *Patidar & Patel, 2022*), approaches for collecting embedded opinions about product and service attributes remain difficult. Future research should focus on improving aspect-based sentiment analysis procedures for e-commerce and tourism, which require a deep understanding of customer opinions. It has been reported that sentiment analysis of different internet user data is challenging and requires creative methods (*Bansal & Srivastava, 2019*; *Karthikayini & Srinath, 2018*). Advanced natural NLP approaches, such as those described by *Patidar & Patel (2022)*, may improve accuracy and applicability of aspect-based sentiment analysis across different domains. As suggested by *Chen & Yao (2021)*, data pre-processing and categorization ambiguity must also be addressed to better understand customer feedback. Moreover, user views can change over time due to various factors such as evolving trends, personal experiences, or external influences. Therefore, future aspect-based sentiment analysis models must be equipped to adapt to these changing complexities to better grasp customer sentiments across different scenarios.

*Handling multimodal data*: Since online reviews increasingly include visual information, multimodal techniques that analyze text and images are crucial. Previous research (*Lee et al., 2022*; *Limsopatham, Rokhlenko & Carmel, 2018*) have recognized the usefulness of multimodal techniques in analysis of online customer reviews. These methods go beyond text analysis to picture sentiment analysis. Multimodal data is important, but various problems in handling data from different sources with different formats still need to be solved. Effective algorithms blending textual and visual data smoothly to obtain sentiment insights are challenging. Image sentiment analysis requires novel methods to fully understand emotional complexities of visual content. Standardized benchmarks and assessment measures for multimodal sentiment analysis are needed to ensure model performance. In the ever-changing world of online reviews, overcoming these problems will be essential to maximize multimodal techniques and improve sentiment analysis.

*Dealing with sarcasm and irony*: More complex algorithms that can recognize and analyze sarcasm and irony in customer reviews are needed to improve analysis of online reviews. Sarcasm and irony are difficult to identify and grasp, even using sentiment

classification methods. *Maheswari & Dhenakaran (2023)* introduced a technique for spotting sarcastic thoughts in online communication, emphasizing the necessity for novel approaches to capture every aspect of language. *Clavel Quintero & Arco García (2018)* used character language model classifiers to identify irony in social media and e-commerce communications. The study shows how difficult it is to spot irony in user-generated content on Twitter and e-commerce sites. Additionally, *Al-Ghuribi, Noah & Mohammed (2023)* presented CF approaches using sentiment analysis on user reviews, illustrating the persistent problem of managing metaphorical language. Future sentiment analysis research must refine existing models and explore new methods to better understand sarcasm and irony in customer reviews, improving system accuracy.

*Fake review detection*: Even with advances in this field, maintaining the correctness of these models is vital, especially given the dynamic and changing nature of online deception. Existing studies have highlighted the need for new methods and improvements to detect fake reviews (*Anas & Kumari, 2021*; *Birim et al., 2022*; *Jin, Zhang & Zhang, 2019*). Because of the flexibility of fraudsters, it is necessary to investigate new ways to keep up with evolving misleading practices. Future research should use state-of-the-art machine learning, deep learning, and NLP techniques to identify more complex fake reviews. Moreover, researchers, platforms, and regulatory agencies must collaborate to build a viable fake review detection methodology. These issues must be addressed to preserve online reviews and build the confidence of customers and businesses.

*User-generated content challenges*: Existing research on user-generated content problems reveals crucial factors that need additional focus on the improvement of review analysis models. Importantly, these models must be more adaptable to different language settings. Existing studies emphasize the need to address language, writing, and cultural differences. To make sentiment analysis systems more robust and effective, methods must accommodate linguistic complexities for user-generated content (*Ravichandran & Deng, 2023*). This requires methods that can detect sentiment expressions in different languages, accommodate different writing styles, and delicately capture distinctive cultural aspects in the text. Overcoming these problems will help sentiment analysis algorithms become more applicable and reliable in varied language and cultural situations as user-generated content evolves. The literature provides vital insights into existing attempts to address these difficulties and establish the framework for future initiatives.

*Integration of machine learning models*: Incorporating machine learning models in sentiment analysis presents challenges and research opportunities. Despite advances in this field, thorough studies that assess the performance of different machine learning algorithms across different contexts are needed. Understanding the strengths and weaknesses of different models is essential to finding the best solutions for different applications (*Araque et al., 2017*). Future research should overcome this gap for advanced review analysis by comparing performance of different machine learning models. Such investigations can improve models and provide new methods, advancing the field of study. The adaptability and scalability of these models for multiple datasets and domains must be studied for implementation and real-world application. As sentiment analysis evolves,

improving machine learning model integration will improve the accuracy of review classification and reliability across various contexts.

*Explainable and interpretable models*: Developing explainable and interpretable models is essential for understanding the decision-making process of the models being used for review analysis. Existing studies offer several effective methods to develop explainable and interpretable models (*Linardatos, Papastefanopoulos & Kotsiantis, 2020*). However, federated models may need more interpretability, making their decision-making procedures challenging to understand. The need for more transparency in federated learning models raises questions about their dependability and trustworthiness in real-world applications. Federated learning interpretability research is needed to solve this problem. Understanding the decision processes of federated sentiment analysis models will improve their practicality and promote openness and accountability in machine learning applications. Future research should focus on building methods to understand federated models, allowing stakeholders to understand and accept sentiment analysis results across multiple domains.

*Cross-domain generalization*: Cross-domain generalization strategies must be explored to improve the adaptation of review analysis model across domains. Existing review analysis methods have been successful in e-commerce, tourism, and social media, but generalizing them to other domains is difficult. The findings suggest ways to improve review analysis model adaptability and performance in new or diversified environments. To achieve robust cross-domain generalization, domain-specific subtleties, language differences, and user expressions must be overcome. Future research might focus on transfer learning, using pre-trained models to capture domain-neutral information and domain adaptation to fine-tune models for specific domains (*Lai et al., 2024*). Addressing cross-domain generalization difficulties makes review analysis models more versatile and functional in real-world settings with different and dynamic domains.

*Real-time sentiment analysis*: Applications that need real-time insights require fast algorithms and models for the analysis of online customer reviews. Recent studies (*Mantilla-Saltos et al., 2023*) have explored real-time sentiment analysis methods, but it is noted that applying more robust algorithms and models is necessary. In applications that need quick insights, sentiment analysis must be fast and accurate. Developing real-time methods to handle and analyze massive data streams and respond to changing user views and contextual details is difficult. Real-time sentiment analysis in customer feedback, social media monitoring, and online reviews requires addressing data scalability, algorithm efficiency, and model adaptation. Novel real-time data stream handling methods, model architecture optimization for quick inference, and sentiment dynamics temporal relationships may be explored in future studies. Integration of edge computing and effective parallel processing might also improve real-time sentiment analysis systems (*Qiao, 2023*).

*Ethical considerations*: To responsibly deploy customer review analysis models, approaches and frameworks must address ethical issues, including algorithm bias, privacy, and consumer data usage. This is stressed by several studies (*Boumhidi & Nfaoui, 2021*; *Sánchez, 2022*) that have contributed to the discussions on the ethics of review analysis

approaches. Algorithm biases must be acknowledged and mitigated to avoid unforeseen outcomes and maintain fairness. Privacy problems, especially in regulated businesses, require rigorous methods and structures. Federated learning prioritizes data protection and addresses privacy challenges. Federated learning improves privacy and ethics by keeping raw data on local devices and only sharing model changes (*Rahman et al., 2023*). Creating and following ethical norms will help to build trust, transparency, and responsible innovation in the analysis of customer reviews.

This study provides a detailed overview of the application of NLP in the analysis of online customer reviews, the challenges and potential future research directions to develop advanced review analysis approaches, and their practical applications. Interdisciplinary research becomes essential to address these issues and improve the analysis of online customer reviews for various applications in this digital era.

## CONCLUSION

This study examines 154 articles published between 2013 and 2023, revealing a decade of NLP advances in online customer review analysis. The literature shows that sentiment analysis approaches have evolved to improve the accuracy and quality of opinion mining. Researchers have successfully used advanced algorithms and machine learning models, from SVM and genetic algorithms to cutting-edge methods like BERT and deep neural networks. The findings suggest that the review analysis is being used in e-commerce, tourism, and other industries to improve product, marketing, and decision-making. In addition, the taxonomy offered in this study provides an organized summary of the changing landscape by classifying the research based on applications. Even though there have been significant improvements in the performance of review analysis and the exploration of new methodologies in the reviewed literature, there are still several open research challenges. These challenges include detecting and preventing fake reviews, integrating multi-modal data for better analysis, transferring sentiment knowledge across different domains, handling diverse data sources, conducting aspect-based sentiment analysis, dealing with sarcasm and irony, addressing user-generated content challenges, integrating machine learning models effectively, developing explainable and interpretable models, achieving cross-domain generalization, ensuring real-time sentiment analysis, and addressing ethical issues. Future work will address these problems to produce more robust and universally applicable NLP models for online customer reviews as the industry evolves. This survey shows how NLP research affects many areas and can change how businesses interpret and exploit customer reviews in the digital age.

### Funding
The authors received no funding for this work.

### Competing Interests
The authors declare that they have no competing interests.

## Author Contributions

- Nadia Malik conceived and designed the experiments, performed the experiments, analyzed the data, performed the computation work, prepared figures and/or tables, authored or reviewed drafts of the article, collecting Papers for Databases and tabulating them, and approved the final draft.
- Muhammad Bilal conceived and designed the experiments, performed the experiments, analyzed the data, performed the computation work, prepared figures and/or tables, authored or reviewed drafts of the article, defined the taxonomy, and approved the final draft.

## Data Availability

This is a literature review.

## Supplemental Information

Supplemental information for this article can be found online at http://dx.doi.org/10.7717/peerj-cs.2203#supplemental-information.

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
