# Peer review of "Natural language processing for analyzing online customer reviews: a survey, taxonomy, and open research challenges"

_PeerJ Computer Science, doi:10.7717/peerj-cs.2203_

## Round 0.1 · original submission · Major Revisions

Experts have now judged your manuscript. You are asked to undertake a major revision in which you carefully take all comments and suggestions of the reviewers into account and answer them in a response letter. This concerns in particular more detailed information on the survey methodology (e.g., search queries and repositories used, also for replication purposes), on the taxonomy (e.g., definition of the different elements and discussion of their intersections), and on some of the challenges (e.g., Aspect-Based Sentiment Analysis needs more explanation). Moreover, the manuscript could benefit from proofreading by a fluent English speaker.

·

Basic reporting

Main considerations:

1. Abstract: Some statements are not right (or too strong): For example, the first sentence in the abstract, "In recent years, e-commerce platforms have replaced conventional marketplaces" is not 100% correct. Although E-commerce platforms have gained significant prominence and market share, they have NOT completely replacing conventional marketplaces. it is better to say e.g. ""In recent years, e-commerce platforms have become popular and transformed the way people buy and sell goods."

2. In Survey Methodology, how the keywords are used to search for papers needs to be explained e.g. when you search them you use A or B or C , alternatively A and B and C etc
3. In Taxonomy section, either rename the figure e.g. remove 'NLP in' or add '... NLP in analysing ...' . 4. The reviewer also suggests to redesign the diagram. This taxonomy is very high level and some items overlap with each other in coverage. It's better to change Customer Feedback and Satisfaction to Customer Satisfaction/user experience enhancement(as customer reviews are customer feedbacks). It's better to change Review Analysis and Management to Fake Review Detection and Management as Review Analysis overlaps with Sentiment Analysis. Also sentiment analysis and user satisfaction overlaps to some extent as the purpose/result of sentiment analysis is to understand user satisfaction. It might be better to change Customer satisfaction to Customer Loyalty The reviewer felt that this Taxonomy is more about possible usage/applications of online customer reviews, rather than what NLP can do in this field. Typical applications of NLP in analysing customer reviews include sentiment analysis, recommendation, topic modeling, entity recognition, fake review detection, summarization, trend analysis etc.
5. In Sentiment Analysis and Opinion Mining subsection, it seems the paragraphs are organised in terms of application areas e.g. tourism (tripAdvisor, Amazon, Mobile ) etc. It is recommended to give each application area a short subsection title to better guide the reader. If the paragraphs are organised in terms of analysis techniques e.g. deep learning, conventional machine learning etc, please group all deep-learning related papers into one paragraph and give it a subsection title. Do the same to group other technologies e.g. GA, Lexicon-based analysis, Word Clouds etc into different paragraphs.
6. In the subsection Customer Feedback and Satisfaction, some content overlaps with the coverage of sentiment analysis. It is better to redesign the coverage of Customer Feedback and Satisfaction and Sentiment Analysis and Opinion Mining subsections, making each subsection cover unique content.
In the Marketing and Brand Management section, it is recommended to add some subsection titles to the main points to guide the readers. This section is very long and contains no subsection titles, no figures, and no tables. The reader can easily get lost in the lengthy text.
In the Discussion section, it is recommended to add some subsection titles to the main points to guide the readers, similar to the ones used in the Open Challenges and Future Directions section.
7. In Discussion, some challenges need more explanations. For example, in Aspect-Based Sentiment Analysis, the ambiguity, user views’ changing subtleties and complexities issues all need more explanations. For example, you may add 'Limitations' before the last paragraph in this Discussion section. You may add a subsection title 'The landscape of analysis techniques' before the paragraph "The changing landscape of techniques ....". You may also draw a diagram to illustrate the related techniques and their relationships.
8. The Open Challenges and Future Directions is well written.
9. In Conclusion, the main open challenges should be summarised and be consistent with those discussed in the Discussion section e.g. real-time, ethical issues, interpretability etc. The inconsistent summary of challenges mentioned in the Conclusion makes it harder to follow. The authors either re-group the challenges mentioned in the Discussion section into larger groups, or list all the challenges in the Conclusion section.
10. NOTE: PeerJ uses the APA ('Name. Year') style with an alphabetized reference list. The format of ALL in-text citations should be changed accordingly.


Minor writing style changes recommended:
In Introduction, I suggest to Remove "their number and textual nature [3]. Traditional manual review analysis is impossible due to" [Note: keep ref [3]] [you have a similar but better sentence in the Aim paragraph]
In Survey Methodology, finish the sentence with full stop '.': "... Fake Reviews were searched"
In Discussion, Instead of "Despite advances like [118] and [148],", list the actual advances then followed by references.
Expand ABSA (the use of the Acronym ABSA here does not save much space but hinders the readability of the paper).
Instead fo saying "[51] analyses", "[71] mines Twitter", list the author name first, then text, then references. E.g. change "[51] analyses" to " Roccabruna, Azzolin, and Riccardi analyse .... [51] "
included in Section 3 => included in Section Taxonomy of ... [as you section titles are not numbered]

Experimental design

The survey methodology is valid overall. There are some rooms for improvements.

Validity of the findings

The findings are valid.

Reviewer 2 ·

Basic reporting

The manuscript needs complete proofreading by a fluent English speaker before being accepted for publication. literature review section may be included in the form of table for few studies in same area.

Experimental design

The manuscript is suitable as per the journal scope.

Validity of the findings

The author presented findings in good form.

Additional comments

The author needs to consider a few comments, as mentioned below:
1) Researchers queries and their solutions should be included in the discussion section.

2) Future scope is missing.

3) Separate discussion section is recommended.

·

Basic reporting

The paper presents a literature survey on Natural Language Processing applied to the problem of online user feedback analysis. The objective is that of specifying a taxonomy of the different applications of NLP for feedback analysis and to identify open research challenges.

The paper is in general clear and well structured. Some aspects should be revised as specified in the general comments below. The literature has been deeply analysed. Figures and tables seem to be sufficiently clear to support the argumentation in the paper. Some terms/concepts used in the paper should be introduced/clarified (see the detailed comments below).

Experimental design

The design of the study have been introduced. Some aspects should be clarified or reported, such as a more precise definition of the queries that have been used in the study (to support the replication of the study) and a more precise definition of the research questions at the basis of the literature survey.

Validity of the findings

Further details should be provided at support of some of the open challenges proposed in the paper.

Additional comments

The paper addresses an interesting problem that is the possibility to perform a deep analysis of the feedback in order to accomplish several objectives such as the feedback sentiment identification or the user profiling for recommendation purposes. Some aspects should be revised and better described.

In the description of the methodology it would be interesting to have information about the sources considered in the work to retrieve the different papers. Specifically if the search queries have been used to search only in scientific databases, such as Scopus, ACM digital library, or the search has been extended to the whole internet.

A first concern refers to taxonomy specifications. The way in which taxonomy has been identified should be better clarified in the initial part of the "Taxonomy of NLP questions in the revisions of online customers". Does taxonomy derive from literature or derives from the hypotheses of the authors who have been used to classify the articles in the literature? Please clarify this point.

Concerning the different elements/concepts of the taxonomy (e.g., Sentiment Analysis and Opinion Mining), it would be beneficial for the reader to provide a definition of each one of them before describing the different related works. This should be a way to clearly declare the meaning of the elements of the taxonomy before illustrating the way the literature deals with those elements. Another aspect concerns the closing of the different sections describing the elements of the taxonomy. It would interesting to have for each one of them a short summary of the most relevant aspects that have been identified in the literature.

One aspect that should also be discussed is the fact that the different elements of the taxonomy appear to have “intersections”; for example, “Sentiment Analysis and Opinion Mining”, appears to be related to “Customer Feedback and Satisfaction”. It would be interesting to provide a discussion on the (possible) relationships between the different elements of the taxonomy.


Minor issues
- Line 73: Hower -> However
- Line 795: briefly introduce the concept of "aspect-based sentiment analysis"

---

## Round 0.2 · accepted · Accept

Please do take the remaining minor reviewer comments into account when preparing the final version in production.

·

Basic reporting

The reviewer is happy with the rebuttal letter and the revision.

Some minor revisions could be made. For example,
1) BERT and Neural Network were helpful: revise this phrase. BERT is a type of Neural Network. You'd better avoid listing these two together as 'a and b'.
2) The reviewer suggests to change "taxonomy of NLP applications highlights" to "taxonomy of NLP applications in online customer review analysis highlights" to be more specific.

Experimental design

The revised survey methodology is comprehensive.
The explicitly listed research questions are clear and helps guide the readers.

Validity of the findings

The findings answered the research questions.

·

Basic reporting

The paper presents a literature survey on Natural Language Processing applied to the problem of online user feedback analysis. The objective is that of specifying a taxonomy of the different applications of NLP for feedback analysis and to identify open research challenges.

Experimental design

The revision of the paper clarified the concerns related to the definition of the queries used in the literature survey and those related to the research questions.

Validity of the findings

The revision clarified the missing aspects of the previous version.

Additional comments

The revised version of the work addressed the concerns reported in the previous review adding several clarifications that explains some of the design choices made in the preparation and analysis of the survey. The paper seems to be an interesting contribution to the journal.